# The Adipose Tissue-Derived Secretome (ADS) in Obesity Uniquely Regulates the Na-Glucose Transporter SGLT1 in Intestinal Epithelial Cells

**DOI:** 10.3390/cells14161241

**Published:** 2025-08-12

**Authors:** Vivian Naa Amua Wellington, Soudamani Singh

**Affiliations:** Department of Biomedical Sciences, Joan C. Edwards School of Medicine, Marshall University, 1 John Marshall Drive, Huntington, WV 25703, USA; wellington3@marshall.edu

**Keywords:** obesity, adipose-tissue-derived secretome, glucose transport, SGLT1

## Abstract

Obesity is a complex chronic inflammatory condition that results from excess fat accumulation. It increases the risk of developing numerous co-morbidities such as Type 2 diabetes mellitus, cardiovascular disease, hypertension, and stroke. The adipose tissue is itself a vital endocrine organ that secretes numerous adipokines, cytokines, and exosomes, which are collectively known as the adipose-derived secretome (ADS). This ADS has been shown to influence and modulate many physiological processes. During obesity, the composition of ADS is altered, which may contribute to the development of obesity-associated diseases. Type-2 diabetes mellitus is one of the most common complications of obesity due to alterations in glucose homeostasis. Glucose absorption occurs via Na-glucose co-transport via SGLT1 at the brush border membrane (BBM) of small intestinal villus cells. This process of transepithelial glucose uptake is the primary method of glucose absorption from diet. However, how ADS mediates the function of SGLT1 is not yet known. This study aims to determine the mechanism of regulation of SGLT1 by ADS in intestinal epithelial cells. We show that ADS from OZR (but not LZR) stimulates SGLT1 in IEC-18 cells. OZR-ADS treatment diminished Na/K-ATPase activity in IEC-18 cells. Kinetic studies indicated that the mechanism of stimulation for SGLT1 during OZR-ADS treatment was secondary to an increase in the affinity (1/K_m_) of the co-transporter for glucose without a change in co-transporter number. Western blot studies revealed that SGLT1 protein expression was unaltered in the two groups, confirming our kinetic studies. Immunoprecipitation demonstrated that an increase in the affinity of the SGLT1 protein was mediated by altered phosphorylation. In conclusion, during obesity, the adipose tissue secretome stimulates SGLT1 in intestinal epithelial cells, leading to an increase in affinity for glucose. The affinity change is due to alterations in SGLT1 phosphorylation. Together, these results may provide important insight into the mechanisms underlying altered glucose homeostasis in obesity and how this may lead to the development of Type 2 diabetes mellitus.

## 1. Introduction

Obesity is a major global health concern [1]. The World Health Organization, which defines obesity as a body mass index (BMI) ≥ 30 kg/m^2^ and overweight as a BMI ≥ 25 kg/m^2^, reported that in 2022, over 2.5 billion adults were overweight, with 890 million of these living with obesity [2,3]. In the United States alone, over 2 in 5 adults (42.4%) are considered obese [4,5]. Another report estimated that around 167 million people would face adverse health effects due to obesity in 2025 [3]. If current trends continue, it is projected that by 2030, 58% of the world’s population will be overweight or obese, including over 85% of Americans [6]. Moreover, the economic costs of overweight and obesity is projected to surpass USD 18 trillion by 2060 if these trends persist [7]. Obesity may be genetic and (or) environmental and arises from an imbalance between caloric intake and energy expenditure, leading to the excessive accumulation of adipose tissue. [3,4,5,8,9]. It is a multifaceted disease influenced by various factors, including diet, genetics, sleep, physical activity, environment, culture, and socioeconomic status [6,10,11].

Obesity plays a significant role in the pathogenesis and progression of numerous chronic conditions, including metabolic syndrome, diabetes mellitus, cardiovascular disease, stroke, and certain cancers [10,12,13]. Obesity is a chronic inflammatory condition and affects nearly every organ system [13]. The associated inflammation is first brought on by the presence of an excess of macronutrients, most of which is stored in the adipose tissue [14]. The adipose tissue is not just a caloric reservoir but also functions as an endocrine organ that secretes a range of hormones, growth factors, adipokines, cytokines, enzymes, and extracellular vesicles [15,16]. The secretory factors released by the adipose tissue, collectively referred to as the “adipose-tissue-derived secretome (ADS)”, exert systemic effects on target tissues and regulate various physiological processes [17,18].

During obesity, the contents of ADS are significantly altered; there is a marked increase in the secretion of the pro-inflammatory adipokines IL-1, IL-6, TNF-α, TNF- β, angiotensin, leptin, visfatin, and resistin, as well as a decrease in the anti-inflammatory adipokines adiponectin and IL-10 [10,12,13,19]. The decrease in adiponectin contributes to a pro-inflammatory state in the body [15,16]. Additionally, these pro-inflammatory mediators can impair the activation of insulin signaling receptors on pancreatic β-cells, promoting the development of insulin resistance [17,18]. The imbalance of adipokines that occurs during obesity contributes to the development of obesity-associated diseases.

Obesity alters glucose homeostasis [20]. Carbohydrates, lipids, and proteins ultimately break down into glucose, which then serves as the primary metabolic fuel for mammals. Emerging evidence suggests that one of the key complications of obesity is the development of insulin resistance, a central component in the pathogenesis of Type 2 diabetes, characterized by elevated blood glucose levels [21]. Consequently, understanding the regulation of glucose homeostasis and its impact on energy metabolism is of utmost importance [22].

Intestinal absorption of glucose plays a critical role in glucose homeostasis [23]. In vivo, the primary route of glucose absorption across the intestinal epithelium involves active transport mediated by the Na^+^-glucose co-transporter (SGLT1), which is located in the brush border membrane (BBM) of enterocytes. SGLT1, which is abundantly expressed in the BBM of intestinal villus cells, facilitates the transport of one glucose molecule coupled with two sodium ions into enterocytes [24,25]. This transporter is crucial for the absorption of sodium and carbohydrates in the normal intestine [26]. However, how glucose absorption may be altered in the context of obesity remains poorly understood.

Physiologically, the absorption of glucose in the small intestine contributes to regulating blood glucose levels and may be a potential target for treating hyperglycemia. Recent advancements in understanding the molecular and cellular mechanisms underlying glucose absorption in the gut and its reabsorption in the kidney have paved the way for novel strategies in diabetes treatment. Furthermore, changes in blood glucose levels are known to impact appetite regulation, suggesting that glucose absorption may be relevant to the hyperphagia observed in metabolic diseases [27].

Previous studies have shown that in diabetes, SGLT1 expression is increased in the small intestine [28]. In addition, Fiorentino et al. recently showed that people with obesity have higher duodenal SGLT-1 and GLUT-5 levels, which are linked to increased postprandial glucose, insulin resistance, and hyperinsulinemia [29]. Factors such as EGF, ATP, and glucagon-like peptide-2 have been shown to upregulate SGLT1 expression, while TNF-α, IFN-γ, luminal leptin, and RS1 (an intracellular 617 amino acid protein) have been found to decrease SGLT1 levels [30]. The regulation of SGLT1 occurs at both the transcriptional and post-translational levels. Transcriptionally, maximal SGLT1 promoter activity relies on the presence of essential binding sites for transcription factors such as Sp1 and HNF1 [31]. At the post-translational level, regulatory enzymes such as SGK1, PKC, and PKA have been implicated in modulating SGLT1 function [30,32,33].

Considering that altered glucose homeostasis is a key cause of diabetes in obesity, this study aims to investigate how the adipose-tissue-derived secretome (ADS) regulates intestinal SGLT1-driven glucose absorption during obesity.

## 2. Materials and Methods

### 2.1. Animal Studies

Obese Zucker rats (OZR) (male [Leprfa] at 18 weeks of age) and lean Zucker rats (LZR) were acquired from Charles River Laboratories International, Inc. (Wilmington, MA, USA). The rats were subjected to a one-week acclimatization period in the animal facility. They were housed in a controlled environment with a 12 h light–dark cycle and provided unrestricted access to food and water. Obese Zucker rats display visible obesity by five weeks of age, with ongoing fat accumulation throughout their lifespan. The small intestine and visceral fat from both LZR and OZR rats were collected for the study. All animal procedures adhered to the guidelines and ethical regulations set forth by Marshall University’s Institutional Animal Care and Use Committee.

### 2.2. Villus Cell Isolation

Villus cells were isolated from the terminal small intestine of lean and obese Zucker rats using a calcium chelation method, as previously described [34]. In brief, a 12-inch section of the distal small intestine was perfused with a buffer solution composed of 0.15 mM EDTA, 112 mM NaCl, 25 mM NaHCO_3_, 2.4 mM K2PO_4_, 0.4 mM KH_2_PO_4_, 2.5 mM L-glutamine, 0.5 mM β-hydroxybutyrate, and 0.5 mM dithiothreitol; pH of 7.4. The buffer was gassed with a mixture of 95% oxygen and 5% carbon dioxide at 37 °C. The intestine was then incubated for 3 min and gently palpated for 3 min to facilitate cell dissociation. The resulting cell suspension was immediately utilized for whole-cell uptake experiments, or alternatively, phenylmethyl sulfonyl fluoride was added, followed by centrifugation at 1000× *g* for 3 min at room temperature. The resulting cellular pellet was promptly frozen in liquid nitrogen and stored at −80 °C. The isolated cells were subsequently employed for brush border membrane vesicle (BBMV) and Western blot assays

### 2.3. Generation of ADS Condition Media

Lean ADS was obtained from the visceral fat of lean Zucker rats (LZR-ADS). Obese ADS was obtained from the visceral fat of obese Zucker rats (OZR-ADS), a rat model of genetic obesity. The adipose tissues isolated from lean and obese Zucker rats were rinsed, cut into equal-sized pieces (~10 mg (2–4 mm^3^) per piece), and incubated in phenol-red-free, serum-free DMEM (DMEM cat log # 31053036) at 37 °C in 5% CO_2_ for 24 h [35]. The resulting culture was filtered through a 25-micron steel mesh filter into 50 mL Nalgene^®^ Rapid-Flow™ (Rochester, NY, USA) Filter Units (0.2 μm PES Membrane). The ADS media was stored at −80 °C until needed for IEC-18 cell treatment. The ADS media was further diluted 1:10 in media before IEC-18 treatment.

### 2.4. BBMV Preparation and Uptake

BBMVs from rat intestinal villus cells were prepared by MgCl2 precipitation and differential centrifugation as previously described [34]. The final BBMV preparation was incubated for 1 h in a Na-free buffer at room temperature. The uptake process was initiated by adding 5 μL BBMV to a tube containing 95 μL of the previously described reaction mixtures and incubating for exactly 90 s. The reaction was then stopped and filtered through 0.45 μm mixed cellulose ester membrane filters (HAWP; Millipore (Billerica, MA, USA)). The radioactivity was then measured by analyzing the dissolved filters using the procedure described above.

### 2.5. Cell Culture

Immortalized nonmalignant rat intestinal epithelial cells (IEC-18) (CRL-1589 American Type Culture Collection) were maintained in Dulbecco’s modified Eagle Medium DMEM (4.5 g/L glucose, 3.7 g/L NaHCO_3_, 2 mM l glutamine, 10% vol/vol bovine fetal serum, 0.02% insulin, and 0.25%-hydroxybutyric acid). The cells were maintained in medium until 3 days post-confluence. These cells were grown as monolayers in 24-well Transwell plates in a humidified environment at 37 °C with 10% CO_2_. At 3 days post-confluence, the cells were treated with ADS-conditioned media obtained from lean and obese Zucker rats for a duration of 24 h and used in experiments.

### 2.6. Glucose Uptake Studies in IEC-18 Cells

Na-glucose co-transport uptake assays were performed as a measure of the transport of the non-metabolizable glucose analog, ^3^H-O-Methyl D-Glucose (^3^H-OMG), as previously described in IEC-18 cells [34]. Treated cells were washed and incubated for 10 min in a Na-HEPES buffer solution (composed of 47 mM KCl, 1 mM MgSO_4_, 1.2 mM KH_2_PO_4_, 20 mM HEPES, 125 mM CaCl_2_, and 130 mM NaCl; pH 7.4; 37 °C). Subsequently, the uptake process was initiated by incubating the cells in a Tris-HEPES-buffered reaction medium (pH 7.4) containing 130 mmol/L NaCl, 10 μCi of ^3^H-O-methyl glucose (3-OMG), and 100 μmol/L 3-OMG for exactly 2 min. This was performed with and without the addition of 1 mM phlorizin, an SGLT1 inhibitor, and 100 μM of cold D-glucose in the reaction buffers. Ice-cold Na-HEPES buffer containing 25 mM D-glucose was used to terminate the reaction. The cells were then processed following previously described methods and measured using a scintillation counter (LS 6500; Beckman Coulter (Brea, CA, USA)).

### 2.7. Na-K-ATPase Measurement

IEC-18 cells were sonicated for 10 s bursts twice and used for this assay. The Na-K-ATPase activity was measured as inorganic phosphate (Pi) release, as described previously [34]. The activity was expressed as nanomoles of Pi released per milligram protein per minute.

### 2.8. Kinetic Studies in IEC-18

Na-dependent ^3^H-OMG uptake assays were performed at 30 s intervals with varying glucose concentrations (cold OMG 0.1, 0.5, 1, 5, 10, 25, 75, and 100 mM), as previously published [34]. Michaelis–Menten kinetics were used to evaluate uptake values, and nonlinear regression data analysis was performed using Prism 7 software (GraphPad, San Diego, CA, USA).

### 2.9. Protein Quantification

Protein levels were quantified using the Bradford method for uptake studies and Western blot analysis. The DC protein assay kit (Bio-Rad (Hercules, CA, USA)) was utilized, and bovine serum albumin (BSA) served as the standard.

### 2.10. Western Blot Studies

Whole-cell and IEC-18 cell homogenates were centrifuged at 8000× *g* for 5 min at 4 °C. The resulting supernatant was transferred to a new tube and subjected to an additional centrifugation step at 13,000× *g* for 30 min at 4 °C. The resulting pellet was then solubilized in RIPA buffer supplemented with Santa Cruz, CA protease inhibitors. The proteins were separated on an 8% polyacrylamide gel and transferred onto a polyvinylidene difluoride (PVDF) membrane. The membrane was blocked using 5% bovine serum albumin (BSA) for 1 h. Primary rabbit polyclonal antibodies against SGLT1 were applied at a dilution of 1:1000 and incubated overnight at 4 °C. Horseradish-peroxidase-conjugated anti-rabbit secondary antibodies were used at a 1:1000 dilution and incubated at room temperature for 1 h. The immobilized proteins were detected using an enhanced chemiluminescence Western Blotting Detecting Reagent, and the luminescence signals were captured and analyzed by densitometry using an image J. The blots were stripped and re-probed with a mouse monoclonal primary antibody against ezrin (Millipore Sigma (Burlington, MA, USA) MAB3822-C) or β-actin (Santa Cruz Biotechnology (Dallas, TX, USA), Sc-47778) at a dilution of 1:1000, which served as an internal control for normalizing the SGLT1 protein levels.

### 2.11. Phosphorylation Studies

The cell lysates were incubated with an anti-SGLT1 antibody at 4 °C overnight in a rotary shaker. Lysates were incubated with protein A/G Plus-Agarose beads (Thermo Fisher (Waltham, MA, USA)) for 2–3 h the following day. Agarose beads were collected by centrifugation, washed four times with PBS, and supplemented with a protease inhibitor cocktail. The immunoprecipitants were collected after adding Laemmli buffer. The resulting immunoprecipitants were separated by SDS-PAGE and transferred onto nitrocellulose membranes. These membranes were probed with phosphoserine/threonine/tyrosine antibodies (Thermo Fisher). Immunoblots were visualized using enhanced chemiluminescence (Pierce™ ECL Plus (Rockford, IL, USA)).

### 2.12. Statistical Analysis

The data are presented as the mean ± standard error; *p*-values were derived by an unpaired *t*-test. A *p*-value of less than 0.05 was considered statistically significant. In cases where multiple time points were examined, a Bonferroni correction was applied to account for multiple hypothesis tests using Prism 7 software (GraphPad). Nonparametric methods were utilized to evaluate the robustness of the datasets.

## 3. Results

### 3.1. Changes in Na-Dependent Glucose Transport in BBMV in Zucker Rats

To investigate alterations in glucose absorption at the brush border membrane, 3H-OMG (3-O-methyl glucose) uptake experiments were performed in brush border membrane vesicles (BBMVs) isolated from the villus cells of both lean and obese Zucker rats. Na-dependent glucose absorption was significantly increased in the BBMVs from OZR compared to LZR (Figure 1; * *p* < 0.05).

### 3.2. Changes in SGLT1 Protein Expression in Zucker Rats

Since increased expression of SGLT1 may be driving the increased glucose absorption in obesity, Western blot analysis was performed. These studies were conducted at the BBM, as SGLT1 is a BBM protein. SGLT1 BBM expression was not significantly altered in both LZR and OZR, as quantified by densitometry (Figure 2A,B; *p* ≥ 0.05). These data indicate that the stimulation of SGLT1 is not due to an increase in the number of transporters.

### 3.3. Phosphorylation Levels of SGLT1 Protein In Vivo

Since SGLT1 protein expression levels remained unchanged, and substrate affinity may be regulated post-translationally, we investigated the phosphorylation levels of SGLT1. Our results show that the phosphorylation levels of SGLT1 were significantly higher in villus cells from obese animals compared to those from lean animals, as quantified by densitometry analysis (Figure 3A,B; * *p* < 0.05).

### 3.4. Effect of ADS on Sodium-Dependent Glucose Uptake in IEC-18 Cells

To better understand the in vivo regulation of SGLT1 in obesity and define the effect of adipose tissue (AT), we treated 3-day post-confluent IEC-18 cells for 24 h with ADS derived from lean and obese Zucker rats. These cells at 0 days post-confluence phenotypically resemble intestinal crypt cells and mature into villus-like cells by day 4 post-confluence. Our premise is that the regulation of SGLT1 is the result of a balance between positive and negative signals from AT. The results showed that SGLT1 activity, defined as Na-sensitive uptake of 3H-OMG, was significantly induced in OZR-ADS-treated IEC-18 cells compared with LZR-ADS-treated cells (Figure 4; * *p* < 0.05).

### 3.5. Effect of ADS on Na-K-ATPase Activity in IEC-18 Cells

To investigate if this increase in Na-sensitive glucose uptake upon OZR-ADS treatment was driven by an increase in Na-K-ATPase activity, we measured Na-K-ATPase activity by assessing inorganic phosphate release. Consistent with previous in vivo observations, these findings demonstrated significant inhibition of Na-K-ATPase activity in cells treated with OZR-ADS compared to those treated with LZR-ADS (Figure 5; * *p* < 0.05).

### 3.6. Effect of ADS on SGLT1 Kinetics in IEC-18 Cells

Next, experiments were conducted to determine the mechanism by which OZR-ADS stimulates SGLT1 activity. To this end, kinetic studies were conducted in IEC-18 cells. The results show that OZR-ADS stimulated SGLT1′s affinity (1/K_m_) for glucose without changing the maximal rate (Vmax) of glucose uptake. The specific kinetic parameters are shown in Table 1.

### 3.7. Effect of ADS on SGLT1 Protein Expression in IEC-18 Cells

To better understand the mechanism of the OZR-ADS-driven increase in Na-sensitive glucose uptake, Western blot analysis was conducted on treated cells. Protein expression of SGLT1 was not significantly altered in IEC-18 cells upon treatment with OZR-ADS compared to controls, as quantified by densitometry (Figure 6A,B).

### 3.8. Effect of ADS on Phosphorylation Levels of SGLT1 Protein Expression in IEC-18 Cells

Further, we aimed to determine whether the increased phosphorylation of SGLT1 drives increased glucose uptake in vitro. Phosphorylation studies showed significantly elevated SGLT1 phosphorylation in IEC-18 cells treated with OZR-ADS (Figure 7A,B; * *p* < 0.05).

## 4. Discussion

Impaired glucose transport is a defining feature of obesity and contributes to the development of Type 2 diabetes. SGLT1, a high-affinity, low-capacity glucose transporter, is essential for the active absorption of glucose and galactose in the intestine and plays a central role in maintaining glucose homeostasis [36]. Several studies have demonstrated that glucose transport mechanisms differ substantially between obese and non-obese individuals [37]. In obese patients, SGLT1 expression and function are often upregulated, leading to enhanced glucose absorption [38]. More recent data show that individuals with overweight and obesity exhibit a progressive rise in duodenal SGLT1 protein levels compared to lean individuals. This elevated expression is associated with postprandial hyperglycemia, insulin resistance, and elevated circulating insulin levels [29]. Similarly, studies with human enteroids derived from obese individuals (models that retain disease-specific features) have shown aberrant carbohydrate absorption and metabolism [37]. Moreover, increased SGLT1 expression in the duodenum has been documented in individuals with Type 2 diabetes. In line with these findings, increased SGLT1 expression has also been reported across several animal models in response to high dietary carbohydrate intake [39,40,41,42].

Altered SGLT1-mediated glucose absorption has also been well-characterized in genetic models of obesity [43]. An earlier study by Corpe et al. [44] showed that in the Zucker diabetic fatty (ZDF) rat, the small intestinal mucosal mass was 60% greater than that of the Zucker lean control (ZLC) rat. However, the expression levels of SGLT1 mRNA and protein in the ZDF and ZLC rats were the same. This is similar to our findings; further, the findings of our in vivo study closely resemble and extend observations made by Balasubramanian et al. in obese Zucker rats and TallyHo mice villus cells, where SGLT1 stimulation was observed both at the level of the BBM as well as in intact cells from the obese intestine [34]. This study demonstrates that factors secreted from obese adipose tissue (obese ADS) stimulate Na^+^-glucose co-transport. This highlights a novel and significant role for the adipose-tissue-derived secretome in modulating SGLT1 function in the context of genetic obesity.

The adipose tissue secretome has emerged as a focus of metabolic research, with a growing number of adipose-derived proteins, metabolites, lipids, and, recently, exosomes being identified over the past decades. These components exert widespread effects on multiple organs and systems throughout the body [35,45,46]. Understanding how adipose-derived signals influence metabolic function is particularly relevant in the context of obesity. In this study, we highlight the critical role of the obese adipose-derived secretome (ADS) in modulating intestinal glucose absorption via SGLT1. While ADS plays an important role in normal physiology, its altered composition in obesity appears to contribute to the impairment of glucose homeostasis, an essential element in the development of obesity-related Type 2 diabetes. Although the present study did not directly identify the molecular components within the obese ADS responsible for SGLT1 modulation, this remains an important area for future investigation. Moreover, leptin, a well-characterized adipokine, has been shown to regulate SGLT1. Interestingly, intestinal SGLT1 expression is significantly reduced in hyperleptinemic mice but remains unchanged in leptin-deficient models, supporting a role for leptin in downregulating SGLT1 [30,43,47,48]. The precise signaling mechanisms by which leptin modulates SGLT1 remain incompletely defined, but evidence from rat enterocytes suggests that leptin can activate both protein kinase A (PKA) and protein kinase C (PKC), potentially influencing SGLT1 function through these pathways [30].

Post-translational modifications such as phosphorylation are key regulatory mechanisms for SGLT1 expression and function. Indeed, SGLT1 contains several consensus phosphorylation sites for PKA and PKC, although the number and location of these sites vary across species. Notably, PKA-mediated upregulation of SGLT1 activity and expression has been documented in multiple species, including humans, rats, rabbits, and sheep. For example, PKA activation has been shown to directly phosphorylate SGLT1 at serine 418 (Ser418) in Chinese hamster ovary cells, resulting in conformational changes that increase the transporter’s affinity for glucose [49]. Furthermore, amino acid sequence analysis of SGLT1 in *Megalobrama amblycephala* revealed seven potential phosphorylation sites, suggesting that multisite phosphorylation may be a conserved mechanism for modulating transporter behavior [42]. In addition, serum and glucocorticoid-regulated kinase 1 (SGK1) may also play a role in this regulatory network. Hyperactivity of SGK1 has been linked not only to excessive epithelial sodium channel (ENaC) activity and hypertension, but also to increased SGLT1 activity, further linking its dysregulation to obesity and metabolic dysregulation [50]. Further exploration of these signaling networks may help identify new targets for restoring glucose homeostasis in obesity and related metabolic diseases.

The sodium-dependent glucose co-transporter requires Na-K-ATPase for optimal activity, as SGLT1 utilizes an electrochemical gradient to transport two sodium ions and one glucose molecule into the cell. Na-K-ATPase was inhibited in the IEC-18 cells treated with obese ADS, demonstrating that an altered Na-extruding capacity was not a factor in the stimulation of SGLT1. Both obese Zucker rats [34] and IEC-18 cells treated with obese ADS inhibited Na-K-ATPase; thus, the stimulation of SGLT1 during obesity is not secondary to the altered Na-extruding capacity of the cells.

In physiological and pathophysiological conditions, the activity of the transporter changes very rapidly; kinetic studies will reveal the mechanism behind how the transporter responds dynamically to these challenges [51]. In this study, the mechanism of stimulation of Na-glucose co-transport in IEC-18 cells treated with obese ADS is due to an altered affinity of co-transporters for glucose without a change in the number of co-transporters for glucose. This result is consistent with the Zucker obese rat data, where the mechanism of stimulation of SGLT1 was secondary to an increase in the affinity of the co-transporter without a change in the number of co-transporters for SGLT1 [34]. In contrast, in pathophysiological conditions like chronic intestinal inflammation, Na-glucose co-transport is inhibited, and the mechanism of inhibition is due to a decrease in the number of transporters for glucose without a change in the affinity of the transporters for glucose [52]. Thus, in pathophysiological conditions like obesity-induced diabetes, the mechanism of SGLT1stimulation may be mediated by ADS.

BBM was isolated and used to evaluate apical SGLT1 protein expression, which suggests that the mechanism of SGLT1 stimulation by obese ADS was not secondary to altered protein expression. In previously published studies with multiple species like Zucker rat, TallyHo mouse, or obese human intestine, SGLT1 protein levels were reported to be unaffected [34]. These findings were further recapitulated with immunofluorescence studies, which revealed that in obese Zucker rats, SGLT1 expression was unaltered [34].

This study demonstrated a significant increase in the phosphorylation of SGLT1, both in vivo in OZR and in OZR ADS-treated IEC-18 cells. This suggests that the increase in glucose absorption is driven by an increase in the affinity of the co-transporter for its substrate. Affinity-mediated transporter changes can occur either via glycosylation or phosphorylation. For example, in inflammatory bowel disease, inhibition of constitutive nitric oxide results in the inhibition of SGLT1 via decreased glucose affinity secondary to altered glycosylation of the SGLT1 protein [52]. Furthermore, in MCF7 breast cancer, L-Type Amino Acid Transporter 1 (LAT1) mediated leucine uptake is stimulated by obese ADS, and this stimulation is due to the altered affinity of LAT1 without a change in the maximal rate [35]. In this study, the altered affinity of SGLT1 is attributed to increased phosphorylation of SGLT1, since rat SGLT1 has many phosphorylation sites [52], one of these sites may be responsible for the altered phosphorylation observed in this study. However, further site-specific mutation studies are warranted to understand the molecular mechanism of SGLT1 regulation.

In obesity, based on preclinical and clinical data, SGLT1 is stimulated [53,54,55]. Adipose tissue plays a significant role in glucose homeostasis; excess fat causes obesity and is generally associated with insulin resistance and hyperglycemia [46,55]. Studies have shown that in morbidly obese patients, there is an increase in glucose absorption in the small intestine. Increased absorption of glucose in obesity leads to Type 2 diabetes [53,56]. In this study, we have shown the link between adipose tissue and the intestinal glucose transporter SGLT1. These data indicate that changes in SGLT1, both in vivo and in vitro, are mediated through adipocytes.

## 5. Conclusions

In obesity, the adipose-derived secretome stimulates SGLT1 in intestinal epithelial cells. The mechanism of SGLT1 stimulation by obese ADS in IEC-18 cells in vitro is identical to that seen in vivo in villus cells from obese Zucker rats. Both in vivo and in vitro, obese ADS inhibited Na-K-ATPase; thus, the stimulation of SGLT1 during obesity was not secondary to the altered Na-extruding capacity of the cells. The adipose-derived secretome likely mediates stimulation of villus cell SGLT1 during obesity.

## Figures and Tables

**Figure 1 cells-14-01241-f001:**
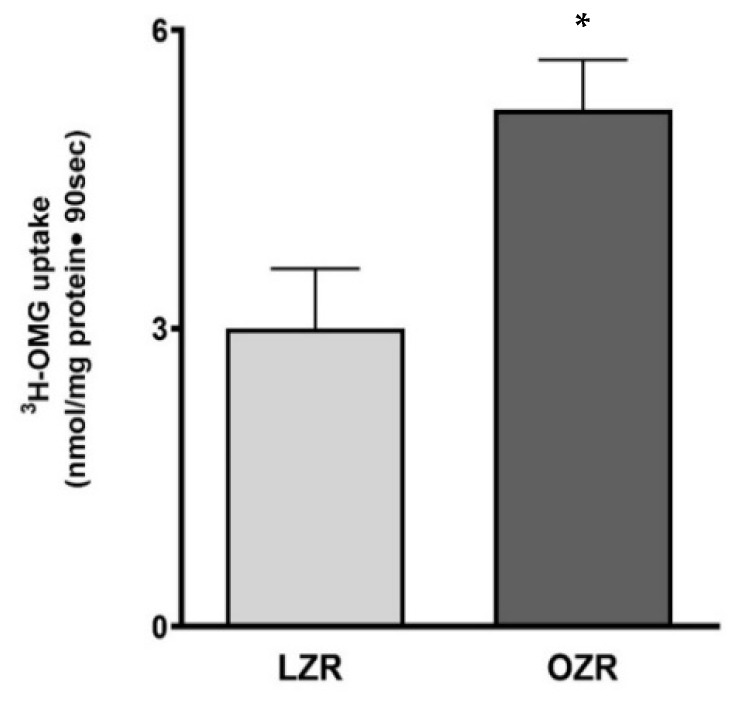
The effect of obesity on SGLT1 activity in villus brush border membrane. SGLT1 was stimulated in brush border membrane vesicle preparations from OZRs compared with LZRs. For all experiments, *n* denotes a unique sample performed with intestinal cells isolated from a separate host. *n* = 4. * *p* < 0.05. The data are shown as the mean ± SEM (error bars).

**Figure 2 cells-14-01241-f002:**
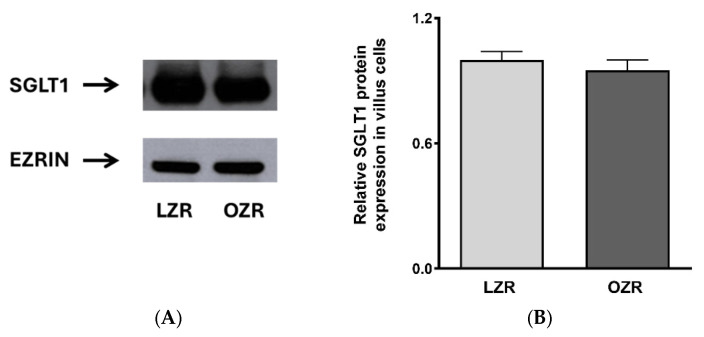
The effect of obesity on SGLT1 protein expression in lean and obese mice. SGLT1 protein levels were unaltered between the groups (**A**,**B**). Experiments were repeated four times (*n* = 4).

**Figure 3 cells-14-01241-f003:**
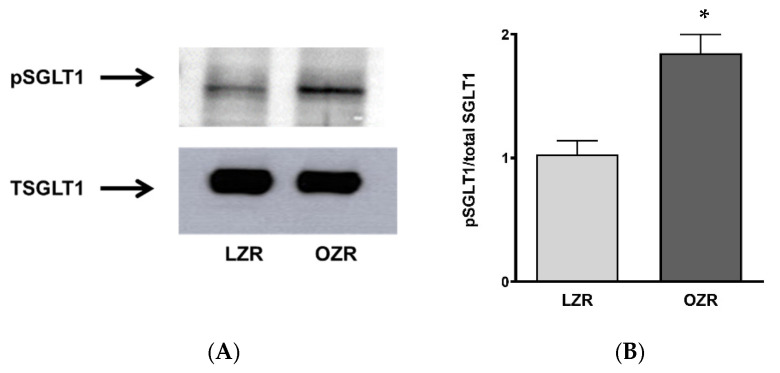
The effect of obesity on pSGLT1 protein expression. The pSGLT1 protein levels were significantly increased in obese compared to lean groups (**A**,**B**). Student’s *T*-test analysis (*n* = 4) indicated by * *p* < 0.05. The data are shown as the mean ± SEM (error bars).

**Figure 4 cells-14-01241-f004:**
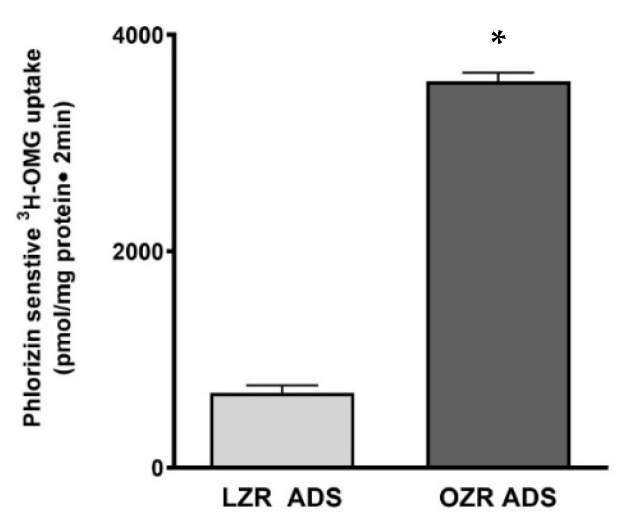
The effect of lean and obese Zucker rat (LZR, OZR) adipose-derived secretome (ADS) on SGLT1 activity in IEC-18 cells. Visceral adipose tissue (AT) was isolated from LZR and OZR. Pieces of AT were cultured for 24 h. After culture, the media were collected and centrifuged to remove tissue debris. The ADS was diluted 1:10 times and applied to IEC-18 cells for 24 h. ^3^H-O-methyl-D-glucose uptake was analyzed. Glucose uptake was significantly stimulated in LZR-ADS compared with OZR-ADS. Student’s *T*-test analysis (*n* = 4) indicated by * *p* < 0.05. The data are shown as the mean ± SEM (error bars).

**Figure 5 cells-14-01241-f005:**
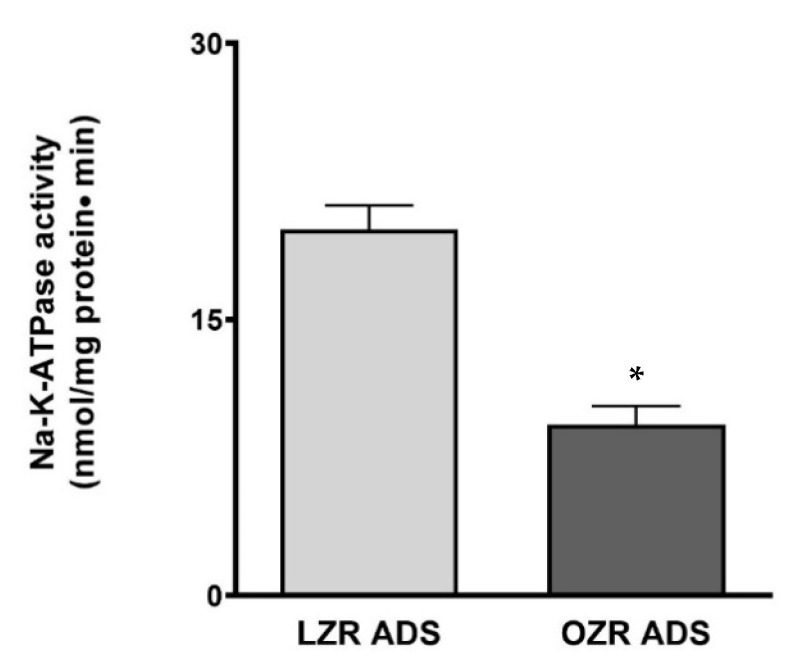
The effect of lean and obese Zucker rat (LZR, OZR) adipose-derived secretome (ADS) on Na-K-ATPase activity. OZR-ADS significantly decreased Na-K-ATPase activity in IEC-18 cells. The Na-K-ATPase activity was determined as a function of Pi release. Student’s *T*-test analysis (*n* = 4) indicated by * *p* < 0.05. The data are shown as the mean ± SEM (error bars).

**Figure 6 cells-14-01241-f006:**
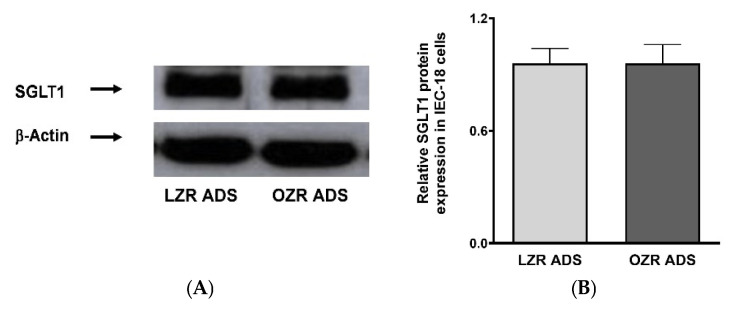
The effect of lean and obese Zucker rat (LZR, OZR) adipose-derived secretome (ADS) on SGLT1 protein expression. The SGLT1 protein levels were unaltered lean LZR-ADS and obese OZR-ADS (**A**,**B**). Experiments were repeated four times (*n* = 4).

**Figure 7 cells-14-01241-f007:**
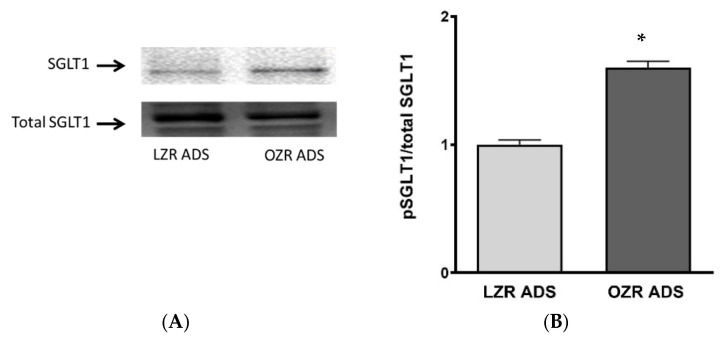
The effect of lean and obese Zucker rat (LZR, OZR) on pSGLT1 protein expression. The pSGLT1 protein levels were significantly increased in obese compared to lean groups (**A**,**B**). Student’s *T*-test analysis (*n* = 4) indicated by * *p* < 0.05. The data are shown as the mean ± SEM (error bars).

**Table 1 cells-14-01241-t001:** The effect of lean and obese Zucker rat (LZR, OZR) adipose-derived secretome (ADS) on SGLT1 kinetic parameters. Na-dependent glucose uptake was determined by varying glucose concentrations, with phlorizin-sensitive ^3^H-O-methyl-D-glucose uptake. The uptake was conducted at 30 s. Student’s *T*-test analysis (*n* = 3) indicated by * *p* < 0.05. The data are shown as the mean ± SEM.

	**Vmax (nmol/mg Protein.30 sec)**	**K** ** _m_ ** **(mM)**
LZR ADS	4.7 ± 0.1	5.0 ± 0.2
OZR ADS	4.9 ± 0.03	2.1 ± 0.1 *

## Data Availability

The datasets used and/or analyzed during the current study are available in this manuscript.

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
