# Peer review of "The Adipose Tissue-Derived Secretome (ADS) in Obesity Uniquely Regulates the Na-Glucose Transporter SGLT1 in Intestinal Epithelial Cells"

_cells, 2025, doi:10.3390/cells14161241_

Round 1

Reviewer 1 Report

Comments and Suggestions for Authors

The article from Amu Wellington and Sing explore the effect o  Adipose derived Secretome  (ADS) in the intestinal SGLT1, exploring the relationship between obesity and diabetes. They used Zuker rats and IEC 18C cells such as models.

The issue is relevant, considering the high incidence of both diseases. The inclusion of kinetic and post-transcriptional changes in SGLT1 is novel, but the 3H-OMG uptake, expression of SGLT1 by western-blot, activity of Na+/K+ ATPase were published by Palaniappan in 2019 (ref 30), being new only the inclusion of IEC 18C cells. Why that data, would be better include more results related. i.e. western blot again specific phosphorylated residues (ser/threo o tyr ) or use of inhibitor of relevant protein kinases in IEC 18C culture .

In introduction, I suggest include the cross-sectional study were found increase in the expresión of SGLT1 in obese (Fiorentino et al, Obesity 2023 DOI: 10.1002/oby.23653)

In results in the figures 4, 5 and 7 the “*” is missed or is displaced, the kinetic assays in table 1 are not included in methods (subitem 2.6)

Evaluate the need to include methodological details in figure legends, since this information might already be provided elsewhere (in methods)

Given that each figure consists of only two panels, it can be improve considering whether related figures should be combined.

In discussion, I suggest fusion lines 362-363 with 376-377, because both are related to Na+/K+ ATPasa (increase or decrease) Other issues

-There misses reference in lines 398-393

- A plausible explication of which meditator in ADS induce SGLT1 phosphorylation

- The author do not discuss why in chronic condition decrease the expression of SGLT1 and in obesity-associated diabetes there is stimulation of the transporter. Again, the article of Fiorentino is not cited

- It would be useful to at least comment on the change in putative residues phosphorylate and its effect described in the literature, or compare it with the other isoform and indicate the similarity percentage of identity between both

Author Response

Reply to reviewer comments cells-3761323:

We sincerely appreciate the time and effort contributed by each Reviewer and Editor. Below, we have carefully detailed the suggested changes to the manuscript.

Reviewer -1

In introduction, I suggest include the cross-sectional study were found increase in the expresión of SGLT1 in obese (Fiorentino et al, Obesity 2023 DOI: 10.1002/oby.23653)

We agree and thank the Reviewer for their suggestion, and we have included Fiorentino et al in the study

In results in the figures 4, 5 and 7 the “*” is missed or is displaced, the kinetic assays in table 1 are not included in methods (subitem 2.6)

Yes, we have included the “*” appropriately in the results, and we have included Table 1 in the methods.

Evaluate the need to include methodological details in figure legends, since this information might already be provided elsewhere (in methods)

Yes, we revised the legend accordingly.

Given that each figure consists of only two panels, it can be improve considering whether related figures should be combined.

We have in vivo and in vitro data; combining the figures could not be feasible. The units on the X-axis differ for each figure, so combining them will not be possible.

In discussion, I suggest fusion lines 362-363 with 376-377, because both are related to Na+/K+ ATPasa (increase or decrease) Other issues

Yes done.

There is a miss reference in lines 398-393

Yes, modified.

A plausible explication of which meditator in ADS induce SGLT1 phosphorylation

We appreciate the reviewer's thoughtful comments. We have analyzed the proteomics data using the ADS; however, we are currently in the process of analyzing it, so we cannot speculate at this time.

The author do not discuss why in chronic condition decrease the expression of SGLT1 and in obesity-associated diabetes there is stimulation of the transporter. Again, the article of Fiorentino is not cited

Thank you for pointing this out. We have now cited this article in our manuscript.

It would be useful to at least comment on the change in putative residues phosphorylate and its effect described in the literature, or compare it with the other isoform and indicate the similarity percentage of identity between both

We thank the Reviewer for their insightful comment. This work is part of an ongoing investigation; we intend to examine the specific phosphorylation site in SGLT-1 in future studies. This is the first study showing the crosstalk between the Adipose tissue-derived secretome and intestinal epithelial transporters.

Reviewer 2 Report

Comments and Suggestions for Authors
  • The authors (Wellington V. et al., The adipose tissue-derived secretome (ADS) in obesity uniquely regulates the Na-glucose transporter SGLT1 in intestinal epithelial cells') performed an experimental study in vivo on the isolation of terminal small intestine cells from obese Zucker rats and in vitro on isolated rat intestinal cells (IEC-18) of ADS (ileum and visceral fat) effects from obese rats. The aim of this study was to understand how ADS could modulate the Na-glucose transporter SGLT1 in isolated intestinal cells and in parallel with a monolayer culture of commercial intestinal cells.
  • The article proposes a cellular enzymatic mechanism for decreased Km of the Na⁺ affinity of this apical membrane transporter induced by ADS incubation. Increased phospholylation of this membrane protein is demonstrated.
  • The paper is clear, and the experimental design is well suited to the hypothesis that ADS decreases the Na⁺ affinity of the intestinal SGLT1 protein by a factor of two.
  • The experimental design consists of obese and lean groups as the appropriate control group.
  • The changes induced by ADS interaction were studied using many complementary methods, such as kinetic analysis with homogenates, western blot analysis of SGLT1 expression, and analysis of protein phosphorylation. An increase in basolateral Na,K-ATPase membrane activity was ruled out as an explanation for the changes in the Na⁺ affinity of the SGLT-1 transporter.
  • It would be helpful to see a figure showing the Na⁺ affinity at different concentrations. Only the Km and Vmax are reported in the table 1.

It would be helpful to include a graphical abstract showing a proposed mechanism for ADS-mediated kinetic regulation of SGLT1 in IEC-18 cells (in both the basolateral and apical cell sides.

Author Response

Reply to reviewer comments cells-3761323:

We sincerely appreciate the time and effort contributed by each Reviewer and Editor. Below, we have carefully detailed the suggested changes to the manuscript.

Reviewer -2

It would be helpful to see a figure showing the Na⁺ affinity at different concentrations. Only the Km and Vmax are reported in Table 1.

The Reviewer is correct in suggesting.  Showing the Michaelis-Menten graph or Lineweaver-Burk plot for the kinetics data is a good choice. However, to compare the differences in Km and Vmax values between the control and obesity groups, we represented them in a simple table format which will be easy to comprehend.

it would be helpful to include a graphical abstract showing a proposed mechanism for ADS-mediated kinetic regulation of SGLT1 in IEC-18 cells (in both the basolateral and apical cell sides.

Thank you for the suggestion. Yes, we now include the graphical abstract.

Reviewer 3 Report

Comments and Suggestions for Authors

Dear Authors,

The manuscript entitled "The Adipose Tissue-Derived Secretome (ADS) in Obesity Uniquely Regulates the Na-glucose transporter SGLT1 in Intestinal Epithelial Cells" is a well-conceived and timely study investigating how obesity-associated ADS modulates intestinal glucose absorption via SGLT1. You provide compelling evidence that OZR-ADS increases SGLT1 activity through post-translational phosphorylation without altering transporter expression or Na-extruding capacity, a novel finding that bridges metabolic and epithelial physiology. The methodology is largely sound, the narrative is clear, and the findings contribute meaningfully to our understanding of glucose homeostasis in obesity.

The comments below only aim at further improving the quality of the manuscript.

  1. Please provide information for specific site(s) on SGLT1 that found to be differentially phosphorylated. Are these serine/threonine or tyrosine residues? The use of general pSer/Thr/Tyr antibodies is too broad. Please provide more explicit details.
  2. Please provide some ELISA data for the profiling of cytokines/adipokines in the ADS from OZR compared to LZR. How do major known cytokines and/o adipokines vary between OZR and LZR?

  3. In Sections 3.3 and 3.8, the p-value is reported as P ≥ 0.05, although you mention there was statistically significant difference between the two conditions. Please check for typographical errors and confirm all statistical annotations. Also, indicate n for each experiment consistently in all figure legends.

  4. The supplementary Western blots lack molecular weight markers. Please provide Western blots with ladder annotations and loading controls for each blot shown.
  5. Please clarify whether the uptake data are normalized to protein content. Present themas nmol/mg protein/min consistently.

Sincerely yours.

Author Response

Reply to reviewer comments cells-3761323:

We sincerely appreciate the time and effort contributed by each Reviewer and Editor. Below, we have carefully detailed the suggested changes to the manuscript.

Reviewer -3

Please provide information for specific site(s) on SGLT1 that found to be differentially phosphorylated. Are these serine/threonine or tyrosine residues? The use of general pSer/Thr/Tyr antibodies is too broad. Please provide more explicit details.

We thank the Reviewer for their very insightful comments. This is a pilot study and part of an ongoing investigation. We plan to focus on the serine/threonine or tyrosine phosphorylation site in SGLT1 in future studies.

Please provide some ELISA data for the profiling of cytokines/adipokines in the ADS from OZR compared to LZR. How do the major known cytokines and/o adipokines vary between OZR and LZR?

We have not performed the cytokine or adipokine ELISA with the ADS at this time. However, this is an ongoing investigation, so we intend to conduct this study in the future using exosomes.

In Sections 3.3 and 3.8, the p-value is reported as P ≥ 0.05,although you mention there was a statistically significant difference between the two conditions. Please check for typographical errors and confirm all statistical annotations. Also, indicate n for each experiment consistently in all figure legends.

We apologize to the Reviewer for the confusion. We agree with and appreciate the Reviewer’s comment and have made modifications to enhance clarity.

The supplementary Western blots lack molecular weight markers. Please provide Western blots with ladder annotations and loading controls for each blot shown.

We apologize for the confusion. In the supplements, we have provided loading controls for the brush border membrane (BBM); ezrin serves as a loading control for BBM. While in IEC-18 cells, we use beta-actin as a loading control.

Please clarify whether the uptake data are normalized to protein content. Present them as nmol/mg protein/min consistently.

Yes, we always normalize the protein content for all uptake data. We typically express it as nmol/mg protein/90 sec for BBM uptake and as pmol/mg protein/2 min for intestinal epithelial cells.

Round 2

Reviewer 1 Report

Comments and Suggestions for Authors

I appreciate your answers to my comments, however I still have some suggestions

1- The authors talk about "glucose transport" when the the determination is of 3-OMG. Both carbohydrates are transported by SGLT- GLUT, but in methology and figure legend could be better use 3-OMG transport or glucose-analog.

2- The kinetic study are not well described. The methodology indicate several glucose concentrations and time of the assays, but which radioactivity isotope and its concentration is not indicate (used glucose, Na or OMG radioactive?)

3- I can understand altho the authors are already working in identify the phosphorylated resides in SGLT1, but  in this new version still the discussion of those putative residues is missed. Only Arthur et al are referenced  to PKC, PKA and PKG (https://doi.org/10.1016/j.bbamem.2014.01.002) but the putative sites or kinases involves are not discussed. Other articles also discuss SGLT1 phosporylation sites.  Han et al. discuss overall SGLT structure https://doi.org/10.1038/s41586-021-04211-w,  Liang et al shown 7 putative phosphorylation sites in blunt snout bream SGLT1 (https://doi.org/10.1038/s41598-021-93534-9), Klinger  et al. in 2018 identified pSer 418 in pig small intestine (doi: 10.14814/phy2.13562)

Author Response

We sincerely appreciate the time and effort contributed by each Reviewer and Editor. Below, we have carefully detailed the suggested changes to the manuscript.

Reviewer-1

1- The authors talk about "glucose transport" when the the determination is of 3-OMG. Both carbohydrates are transported by SGLT- GLUT, but in methology and figure legend could be better use 3-OMG transport or glucose-analog.

Response - Thank you for your observation. We have now addressed this concern.

2- The kinetic study are not well described. The methodology indicate several glucose concentrations and time of the assays, but which radioactivity isotope and its concentration is not indicate (used glucose, Na or OMG radioactive?)

Response - Thank you for your observation. We have now addressed this concern.

3- I can understand although the authors are already working in identify the phosphorylated resides in SGLT1, but  in this new version still the discussion of those putative residues is missed. Only Arthur et al are referenced  to PKC, PKA and PKG (https://doi.org/10.1016/j.bbamem.2014.01.002) but the putative sites or kinases involves are not discussed. Other articles also discuss SGLT1 phosporylation sites.  Han et al. discuss overall SGLT structure https://doi.org/10.1038/s41586-021-04211-w,  Liang et al shown 7 putative phosphorylation sites in blunt snout bream SGLT1 (https://doi.org/10.1038/s41598-021-93534-9), Klinger  et al. in 2018 identified pSer 418 in pig small intestine (doi: 10.14814/phy2.13562)

Response – We have expanded the discussion section to better address our phosphorylation results. Thank you for providing these additional references.

Thank you again for your contribution and your clear, collegial responses.

Reviewer 3 Report

Comments and Suggestions for Authors

Dear Authors,

Thank you for your thoughtful and detailed response to my comments, and for your efforts to revise and clarify the manuscript.

I appreciate the clarifications provided in your rebuttal regarding:

1. The intent to investigate specific phosphorylation sites in follow-up studies;

2. The future inclusion of cytokine/adipokine profiling of the ADS, potentially via exosome analysis;

3. The correction of typographical errors in P-value reporting and inclusion of experimental n in figure legends;

4. The clarification and improved presentation of Western blot data, including the use of appropriate loading controls;

5. The normalization of uptake data to protein content and consistent unit reporting.

While these mechanistic elements remain to be further explored, your transparent acknowledgment of these as current limitations within a pilot study framework is appreciated and scientifically appropriate.

To further strengthen the manuscript, I would encourage that the Discussion section clearly acknowledges:

1. That the specific phosphorylation sites and kinases involved in SGLT1 regulation remain to be identified;

2. That the composition of the ADS was not characterized in this study and will be addressed in future work;

3. The exploratory nature of the findings and the value they offer in generating hypotheses for future mechanistic and translational research.

Thank you again for your contribution and your clear, collegial responses.

Best regards.

Author Response

We sincerely appreciate the time and effort contributed by each Reviewer and Editor. Below, we have carefully detailed the suggested changes to the manuscript

Reviewer 2

I appreciate the clarifications provided in your rebuttal regarding:

  1. The correction of typographical errors in P-value reporting and inclusion of experimental nin figure legends.

Response - Thank you. We have made the necessary changes.

  1. The clarification and improved presentation of Western blot data, including the use of appropriate loading controls

Response - Thank you. We have used Ezrin for BBM samples and Beta-actin for IEC-18 samples, while for phosphorylated SGLT1, total SGLT1 protein was used as a loading control.

  1. To further strengthen the manuscript, I would encourage that the Discussion section clearly acknowledges: That the specific phosphorylation sites and kinases involved in SGLT1 regulation remain to be identified. That the composition of the ADS was not characterized in this study and will be addressed in future work. The exploratory nature of the findings and the value they offer in generating hypotheses for future mechanistic and translational research.

Response - Thank you. We have included it in the discussion.

Thank you again for your contribution and your clear, collegial responses.

Best regards.